# Motion-Focused Tokenization for Source-Free Video Domain Adaptation

**Tzu Ling Liu** [1]  **Ian Stavness** [1]  **Mrigank Rochan** [1]

## Abstract

Source-free video unsupervised domain adaptation (SFVUDA) represents a significant challenge in action recognition research. It requires adapting a pretrained model from a labeled source domain to an unlabeled target domain, with the constraint that source data remains inaccessible during adaptation. Despite advances in SFVUDA approaches, their performance remains significantly inferior to that of the supervised approach. We argue that a key reason for this performance bottleneck is the presence of variable static backgrounds in videos, which contribute substantially to domain shifts. To address this, we propose Motion-Focused Tokenization (MFT) for SFVUDA. In MFT, we first tokenize source and target video frames into patch tokens, then suppress the low-motion tokens, which largely belong to the background, while retaining the motion-rich tokens corresponding to actions for domain adaptation. Experiments introducing MFT to the best-performing existing SFVUDA method demonstrate a significant improvement ($\sim$2%) in its performance across two popular domain adaptation (DA) benchmarks, Daily-DA and UCF-HMDB, covering 15 different DA settings.

## 1. Introduction

Efficiently transferring models across different domains remains a significant challenge in video action recognition. To bridge this gap, Video Unsupervised Domain Adaptation (VUDA) has been proposed, which leverages labeled source domain videos to align feature representations with unlabeled target domain videos (Yang et al., 2020a; Xu et al., 2022a; da Costa et al., 2022; Sahoo et al., 2021). However, in real-world scenarios, direct access to source videos is often restricted due to privacy concerns or data-sharing limitations. To overcome this issue, Source-Free VUDA (SFVUDA) has emerged as an alternative, where the adaptation process relies on a pretrained source model without accessing the source data during adaptation.

Previous SFVUDA methods (Xu et al., 2022b; 2024; Li et al., 2023; Zara et al., 2023) focus on temporal consistency or robust pseudo-labeling methods to mitigate domain shifts. Despite the success of video domain adaptation (DA) methods, they still underperform compared to fully supervised approaches on the target domain, limiting their real-world applicability. We argue that a key bottleneck lies in the presence of low-motion, static backgrounds across both source and target videos. In such cases, models often rely on background appearance rather than motion dynamics, leading to poor generalization. For example, the same action, such as running, may occur in indoor and outdoor settings with vastly different background contexts. These variations introduce significant domain shifts that interfere with the transfer of motion-centric domain-invariant action semantics, which are crucial for effective DA in action recognition.

To address this challenge, we propose the Motion-Focused Tokenization (MFT) module for video domain adaptation, specifically aimed at enhancing SFVUDA. MFT aims to explicitly prioritize regions within video frames that exhibit meaningful motion dynamics. It begins by partitioning both source and target video frames into patch-level tokens. Low-motion tokens, often corresponding to static or redundant background content, are suppressed, while high-motion tokens, which encapsulate key action-related semantics, are retained. This selective emphasis helps reduce background-induced domain shifts and reinforces the model's focus on transferable motion cues essential for action recognition.

MFT offers two key advantages that are particularly beneficial for domain adaptation in video representation learning. First, by selectively enhancing salient motion cues, it ensures that the learned representations emphasize rich and dynamic information crucial for capturing temporal action patterns. Second, by suppressing low motion tokens that are typically associated with static and domain specific background content, MFT mitigates background induced biases that contribute to domain shift. This dual focus on motion enhancement and background suppression leads to more robust and domain invariant video embeddings, ultimately improving generalization across diverse video domains.

In summary, our contributions are: (i) We introduce Motion-Focused Tokenization (MFT), a new module for video do-

---

[1]Department of Computer Science, University of Saskatchewan, Canada. Correspondence to: Tzu Ling Liu <ywa826@usask.ca>.

*Non-archival presentation at ICML 2025 Tokenization Workshop (TokShop)*, Vancouver, Canada. 2025.

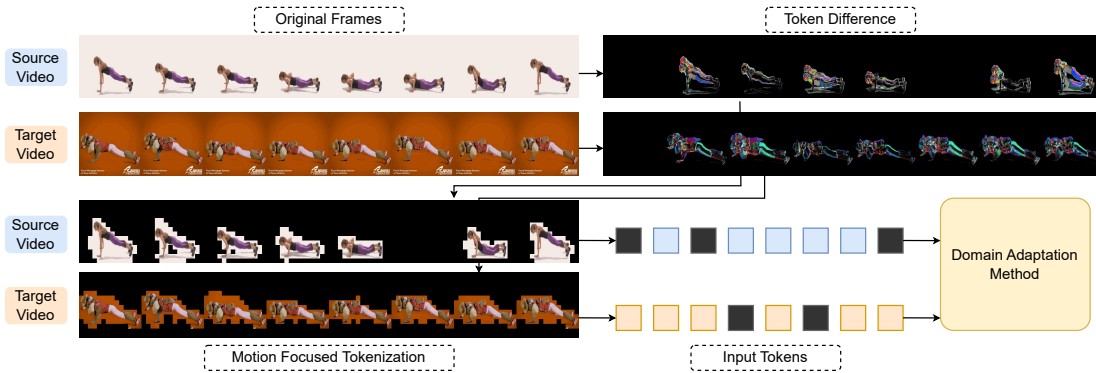

Figure 1: Overview of MFT. For both source and target videos (e.g., a pushing action), MFT tokenizes the frames into patch tokens, computes L1 distance between consecutive temporal tokens, and suppresses those with differences below a threshold $\tau$, which correspond to static, low-motion background. The remaining motion-rich tokens are used for DA. Note that, for SFVUDA, we apply MFT to source videos, obtaining a new pretrained source model that we then adapt to the target domain.

main adaptation that suppresses low-motion tokens while preserving motion-rich action tokens, thereby reducing static background-induced domain shift. (ii) We show that MFT substantially boosts the DA performance of the strongest existing method on two benchmarks spanning 15 diverse DA settings. (iii) We further compare MFT with an alternative strategy and present qualitative analysis.

## 2. Related Work

**Video Unsupervised Domain Adaptation.** In recent years, VUDA has made rapid progress (Yang et al., 2020a; Sahoo et al., 2021; Xu et al., 2022a; da Costa et al., 2022), yet most methods still rely on direct access to source videos during adaptation, which can be impractical due to privacy restrictions. To address this, SFVUDA techniques have emerged, which adapts a pretrained source model to a new target domain without requiring any source data during adaptation. Early work ATCoN (Xu et al., 2022b) and EXTERN (Xu et al., 2024) exploit temporal consistency and regularization to address the problem. Moreover, STHC (Li et al., 2023) adopts stochastic augmentations with consistency learning, while DALL-V (Zara et al., 2023) utilizes CLIP (Radford et al., 2021) and an adapter for target adaptation. However, these methods take full frames into the frameworks. We argue that this would lead to suboptimal cross-domain generalization, as static scene elements often dominate representations. Thus, we propose to retain motion-rich tokens and suppress low-motion tokens to alleviate domain shift.

**Video Tokenization.** Recent advances in video tokenization have explored various strategies to improve efficiency and effectiveness. VideoMAE (Tong et al., 2022) leverages a masked autoencoder for self-supervised learning. However, their masking strategy is random, potentially preserving static backgrounds and amplifying domain shifts. Token Merging (ToMe) (Bolya et al., 2023) progressively fuses pairs of similar tokens based on the similarity score. However, it primarily considers spatial similarity and does not explicitly model temporal dynamics. RLT (Choudhury et al., 2024) encodes pixel differences between consecutive frames and removes low difference tokens. In this work, we adopt the content-aware idea from RLT (Choudhury et al., 2024) and develop motion-focused tokenization (MFT) to mitigate domain shift for SFVUDA.

## 3. Motion Focused Tokenization (MFT)

We introduce Motion-Focused Tokenization (MFT), a novel module that selectively suppresses low-motion regions and retains motion-rich regions in video frames, yielding more robust representations for cross-domain action recognition (Fig. 1). In MFT, we first tokenize the videos into patch-level tokens. Next, we apply a motion-focused criterion to identify motion-rich tokens, which are then used for DA.

**Tokenization of Videos.** Let $\mathbf{V} \in \mathbb{R}^{T \times C \times H \times W}$ represents a video, where $T$ is the number of frames, C represents channels, and each frame has a spatial resolution of $H \times W$. Following the standard tokenization scheme, we partition the video $\mathbf{V}$ into a set $\mathbf{P}$ of non-overlapping patches of uniform size $p \times p$. Each patch $P \in \mathbb{R}^{t \times C \times p \times p}$ corresponds to a spatial location $(x, y)$ in the frame grid, where $x \in [1, H/p]$ and $y \in [1, W/p]$. Each patch $P$ is then treated as a token corresponding to a distinct spatial location in the video. $P_t$ represents a token in frame $t$-th at a specific spatial location. To identify low-motion and motion-rich tokens, for each token $P$, we compute L1 distance among its each pair of consecutive patches $P_1, P_2, \dots, P_T$. This yields pixel-wise motion differences $\mathbf{D}$ as illustrated in Eq. 1. We then calculate the mean of $\mathbf{D}$ across its $p \times p$ values to obtain the patch-level motion energy $\mathbf{E}_P$.

$$\mathbf{D} = \| P_{1:T} - P_{0:T-1} \|_1 \tag{1}$$

| | Method | Top-1 Accuracy on target domain (%) | | | | | | | | | | | | |
|---|---|---|---|---|---|---|---|---|---|---|---|---|---|---|
| | | K→A | K→H | K→M | M→A | M→H | M→K | H→A | H→M | H→K | A→H | A→M | A→K | Avg. |
| | Source Only | 15.6 | 47.9 | 35.7 | 34.7 | 44.6 | 61.6 | 17.5 | 25.5 | 45.1 | 14.6 | 15.5 | 17.8 | 31.3 |
| ZS | CLIP (ResNet50) (Radford et al., 2021) | **30.5** | 50.0 | 42.2 | 30.5 | 50.0 | 62.9 | **30.5** | 42.2 | 62.9 | 50.0 | 42.2 | 62.9 | 46.4 |
| SFUDA | SFDA (Kim et al., 2020) | 12.6 | 44.9 | 27.5 | 16.0 | 35.2 | 49.2 | 13.1 | 24.2 | 24.9 | 16.3 | 13.2 | 25.2 | 25.2 |
| SFUDA | SHOT (Liang et al., 2020) | 12.0 | 44.6 | 29.5 | 15.3 | 36.7 | 51.0 | 13.6 | 24.2 | 21.2 | 17.1 | 14.0 | 24.3 | 25.3 |
| SFUDA | SHOT++ (Liang et al., 2021) | 12.6 | 40.8 | 28.7 | 14.9 | 41.7 | 46.3 | 16.0 | 22.2 | 33.1 | 15.4 | 12.5 | 21.8 | 24.4 |
| SFUDA | MA (Li et al., 2020) | 12.8 | 45.8 | 30.0 | 17.7 | 37.4 | 53.5 | 12.9 | 25.0 | 22.2 | 16.7 | 15.2 | 24.3 | 26.1 |
| SFUDA | BAIT (Yang et al., 2020b) | 12.7 | 45.7 | 30.0 | 16.9 | 39.6 | 53.0 | 13.6 | 25.5 | 21.2 | 15.7 | 14.5 | 25.5 | 26.2 |
| SFUDA | CPGA (Qiu et al., 2021) | 13.1 | 46.0 | 30.7 | 18.1 | 39.2 | 55.1 | 13.1 | 26.2 | 25.5 | 19.2 | 16.5 | 26.7 | 26.5 |
| SFVUDA | ATCoN (Xu et al., 2022b) | 17.2 | 48.2 | 32.5 | 27.2 | 47.3 | 57.7 | 17.9 | 30.7 | 48.5 | 26.7 | 17.2 | 31.0 | 33.5 |
| SFVUDA | EXTERN (Xu et al., 2024) | 23.9 | 55.8 | 35.2 | 18.1 | 53.7 | 68.1 | 26.2 | 40.7 | 57.6 | 26.2 | 18.2 | 51.4 | 39.6 |
| SFVUDA | STHC (Li et al., 2023) | 15.5 | 48.7 | 34.8 | 18.4 | 56.3 | 76.6 | 13.8 | 39.8 | 50.1 | 44.6 | 27.3 | 44.7 | 39.2 |
| SFVUDA | DALL-V (Zara et al., 2023) | 24.0 | 52.5 | 47.0 | 24.0 | **65.4** | 78.1 | 24.0 | **47.0** | **76.7** | **57.9** | 45.7 | 75.0 | 51.4 |
| SFVUDA | DALL-V† (Zara et al., 2023) | 22.8 | 53.8 | 48.9 | 23.8 | 58.3 | 76.8 | 25.0 | 46.8 | 75.1 | 52.5 | **48.8** | 73.9 | 50.5 |
| SFVUDA | DALL-V†+ MFT (Ours) | 24.4 | **57.5** | **49.3** | **31.3** | 60.4 | **79.4** | 26.4 | **47.0** | 74.5 | 55.8 | 47.3 | **74.6** | **52.3** |
| | Target Only | 26.9 | 70.4 | 61.5 | 26.9 | 70.4 | 88.9 | 26.9 | 61.5 | 88.9 | 70.4 | 61.5 | 88.9 | 61.9 |

Table 1: Impact of MFT on the best SFVUDA method, DALL-V on the ***Daily-DA*** benchmark. **Bold** indicates the best performance, underline denotes the best with the same backbone, and †denotes the results from our run of their public code.

| | Method | Accuracy (%) | | |
|---|---|---|---|---|
| | | H→U | U→H | Avg. |
| | Source Only | 71.6 | 76.1 | 73.8 |
| ZS | CLIP (ResNet50) (Radford et al., 2021) | 81.0 | 86.0 | 83.5 |
| SFUDA | SFDA (Kim et al., 2020) | 69.8 | 75.0 | 72.4 |
| SFUDA | SHOT (Liang et al., 2020) | 74.4 | 74.4 | 74.4 |
| SFUDA | SHOT++ (Liang et al., 2021) | 71.1 | 68.1 | 69.6 |
| SFUDA | MA (Li et al., 2020) | 74.4 | 67.3 | 70.9 |
| SFUDA | BAIT (Yang et al., 2020b) | 75.3 | 76.3 | 75.8 |
| SFUDA | CPGA (Qiu et al., 2021) | 75.8 | 68.1 | 72.0 |
| SFVUDA | ATCoN (Xu et al., 2022b) | 85.3 | 79.7 | 82.5 |
| SFVUDA | EXTERN (Xu et al., 2024) | 91.9 | 88.9 | 90.4 |
| SFVUDA | STHC (Li et al., 2023) | 92.1 | 90.9 | **91.5** |
| SFVUDA | DALL-V (Zara et al., 2023) | **93.1** | 88.9 | 91.0 |
| SFVUDA | DALL-V† (Zara et al., 2023) | 88.4 | 90.8 | 89.6 |
| SFVUDA | DALL-V†+ MFT (Ours) | 91.1 | **91.9** | **91.5** |
| | Target Only | 93.7 | 91.4 | 92.6 |

Table 2: Impact of MFT on the existing best SFVUDA method, DALL-V, on the **UCF-HMDB**$_{full}$ benchmark. **Bold** indicates the best performance, while underline represents the best with the same backbone.

Since the first frame lacks a preceding frame for temporal differencing and retaining its full context risks introducing static background domain shift, we approximate its motion energy using the strongest motion energy observed elsewhere in the video. Concretely, we take the maximum motion energy over the temporal index $t$ of the motion energy $\mathbf{E}_P$ to obtain the first frame motion energy $\mathbf{E}_P^{first} = \max_t(\mathbf{E}_P[:, t, :, :])$, where $\max_t(.)$ is applied over the $(T-1)$ temporal dimension. This ensures that $\mathbf{E}_P^{first}$ captures the most salient motion cues in the video. Finally, we concatenate $\mathbf{E}_P^{first}$ with $\mathbf{E}_P$ along the temporal axis to obtain the complete motion energy representation $\mathbf{E}_P^{full}$ for each video:

$$\mathbf{E}_P^{full} = \text{concat}([\mathbf{E}_P^{first}, \mathbf{E}_P]) \qquad (2)$$

By combining $\mathbf{E}_P^{full}$ across every $P$ in the video, we obtain $\mathbf{E}^{full}$ that contains the motion energy values for each token.

**Motion-Focused Tokens for Video Domain Adaptation.** Next, we upsample $\mathbf{E}^{full}$ with resolution $\frac{H}{p} \times \frac{W}{p}$ to the original resolution of the video $H \times W$ using nearest-neighbor interpolation to obtain $\mathbf{E}_{full}^{up}$. We then apply the motion threshold $\tau$ to obtain the final motion mask $\mathbf{M}$:

$$\mathbf{M} = (\mathbf{E}_{full}^{up}) > \tau \in \{0, 1\} \qquad (3)$$

where $\tau$ is a tunable hyperparameter that balances the capture of relevant motion patterns with the exclusion of redundant static tokens. Rather than removing low-motion tokens, which would alter the expected fixed token shape, we suppress the low-motion tokens by setting their values to zero, yielding the final motion-focused video $\mathbf{V}_{mft} = \mathbf{V} \circ \mathbf{M}$, where "$\circ$" denotes element-wise multiplication. We introduce our MFT module on top of the state-of-the-art SFVUDA method, DALL-V (Zara et al., 2023). Specifically, the motion-focused video $\mathbf{V}_{mft}$ is partitioned into motion-focused patches $\mathbf{P}_{mft}$, which are subsequently processed by the ViT-based (Dosovitskiy et al., 2020) vision encoder of DALL-V. For method details, we refer the readers to the DALL-V paper. By utilizing motion-focused action regions identified through MFT, the method is able to improve generalization across diverse domains.

## 4. Experiments

**Datasets and Implementation.** We conduct experiments on two popular VUDA benchmarks: ***Daily-DA*** and ***UCF-HMDB***$_{full}$. *Daily-DA* consists of 18,949 videos drawn from four datasets: ARID (A) (Xu et al., 2021), HMDB51 (H) (Kuehne et al., 2011), Moments-in-Time (M) (Monfort et al., 2019) and Kinetics-600 (K) (Kay et al., 2017), covering eight overlapping categories of daily activities. Note that

| Method | Accuracy (%) | | | | |
|---|---|---|---|---|---|
| | Any→A | Any→H | Any→M | Any→K | Avg. |
| DALL-V[†] | 23.8 | 54.8 | **48.2** | 75.3 | 50.5 |
| DALL-V[†] + RM | 26.1 [+2.3] | 47.7 [-7.1] | 42.1 [-6.1] | 67.6 [-7.7] | 45.9 [-4.6] |
| DALL-V[†] + MFT (Ours) | **27.4** [+3.6] | **57.9** [+3.1] | 47.9 [-0.3] | **76.2** [+0.9] | **52.3** [+1.8] |

Table 3: Comparison between MFT (Ours) and random masking (RM).

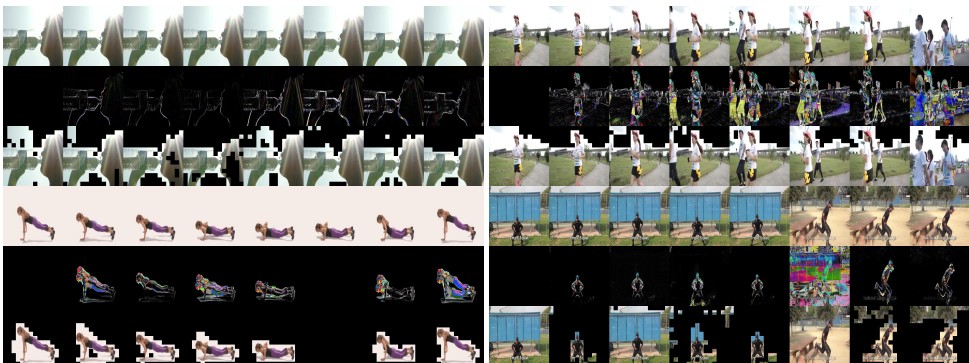

Figure 2: MFT visualization on four videos (two left, two right). Black patches mark static regions. Each video has three rows: original frames, motion differences, and masked frames after MFT. MFT highlights action-related regions, reducing background noise for effective DA.

ARID was filmed under low-light conditions, adding an extra difficulty to the DA task. *UCF-HMDB_{full}* consists of 3,209 videos spanning 12 action classes from the HMDB51 (H) (Kuehne et al., 2011) and UCF101 (U) (Soomro et al., 2012) datasets. We evaluate the effectiveness of our MFT module on the state-of-the-art DA method, DALL-V (Zara et al., 2023). We set $\tau$ in Eq. 3 to 0.005.

**Main Results and Analysis.** In Table 1, we present the results of incorporating MFT into the best existing SFVUDA method, DALL-V, on the *Daily-DA* benchmark. Our MFT module improves DALL-V's performance by an average of 1.8%. Additionally, we report evaluation results on the *UCF-HMDB_{full}* dataset in Table 2, where MFT improves DALL-V by 1.9%. On both benchmarks, our MFT module consistently enhances DALL-V's performance, establishing a new state-of-the-art and demonstrating its effectiveness across diverse DA settings. A performance improvement of approximately 2% is a significant boost for domain adaptation and brings the method closer to the upper-bound supervised Target Only baseline.

**MFT vs. Random Masking.** To evaluate the effectiveness of MFT, we perform an additional experiment on *Daily-DA*, replacing MFT with random masking (RM). In RM, we maintain the same proportion of tokens suppressed as in MFT but randomly mask tokens in each frame, disregarding content and motion cues. Our MFT module consistently outperforms RM (Table 3), demonstrating its superior ability to leverage both content and motion information for improving robustness to domain shifts.

**Qualitative Analysis.** In Fig. 2, we qualitatively assess the effectiveness of MFT by masking low-motion regions with black patches. In the first two examples (left), MFT successfully suppresses irrelevant, domain-variant backgrounds, isolating motion-rich regions. This selective focus on dynamic regions enables robust domain adaptation by prioritizing motion-relevant features corresponding to the action that are invariant across domains, effectively reducing the impact of domain-specific noise or static contextual variations. In the latter two examples (right), where the viewpoint undergoes significant shifts, MFT again consistently preserves the dynamic, motion-rich regions, ensuring reliable tracking of action-centric motion patterns despite changes in perspective or scene composition. By emphasizing action-relevant motion-rich regions over static or domain-specific features, MFT enhances cross-domain generalization, enabling models to adapt seamlessly to new environments.

## 5. Conclusion

We proposed Motion-Focused Tokenization (MFT), a simple yet effective module that prioritizes motion-relevant action content while suppressing variable static background regions, thereby reducing domain shift and improving SFVUDA. Experimental results on two video DA benchmarks showed that the introduction of MFT significantly enhances DA performance. Future work will focus on applying MFT to other video DA models and exploring its potential in unsupervised and semi-supervised video DA.

**Acknowledgements**: We acknowledge the support of the University of Saskatchewan and the Natural Sciences and Engineering Research Council of Canada (NSERC).

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
