# OpenReview forum: "Motion-Focused Tokenization for Source-Free Video Domain Adaptation"
_ICML.cc/2025/Workshop/TokShop — TokShop_

### Official Review · Reviewer_p17P · 2025-06-04

**Rating:** 8
**Confidence:** 4

**Review:**

The paper introduces a video tokenization for domain adaptation that leverages motion information, as a result the proposed algorithm can focus its attention on the relevant spatiotemporal segments of the video. Validated on two benchmarks, Daily-DA and UCF-HMDB, the approach shows strong performance.

**Strengths**: The paper is very well written and easy to follow. The proposed approach is reasonable, simple and seems to be performing well. The validation of the method looks solid and the paper contains a nice mix of qualitative and quantitative results.

**Weaknesses**: It would be nice to add more discussion on methods limitation, e.g., by looking at table one, the method overall looks good, however, on some tasks it underperforms. It would be interesting to discuss this aspect a bit more.

**Questions**:
- Are there any computational gains of MFT vs vanilla tokenization?
- How about different applications of MFT that would go beyond domain adaptation?
- Ablation of tau from eq (1) would strengthen the paper.
- Many reported accuracies are still below 50%, what is holding the UDA? Would even better tokenization (beyond what was proposed in this paper) get us to significantly better results?

---

### Official Review · Reviewer_N9t1 · 2025-06-08
**Review of Motion-Focused Tokenization**

**Rating:** 7
**Confidence:** 3

**Review:**

This paper aims to improve methods in source-free video unsupervised domain adaptation (SFVUDA). Specifically, the authors propose Motion-Focused Tokenization that tokenizes source and target video frames into patches and then retains patches with enough motions (by comparing those in previous frames and next frames). This proposed MFT works on the SOTA SFVUDA method (DALL-V) and improves the domain adaptation performance as suggested by two video DA benchmarks.

---

### Decision · Program_Chairs · 2025-06-10

Accept